# Cell-Free DNA in Plasma and Serum Indicates Disease Severity and Prognosis in Blunt Trauma Patients

**DOI:** 10.3390/diagnostics13061150

**Published:** 2023-03-17

**Authors:** Inga Trulson, Juliane Stahl, Stefan Margraf, Martin Scholz, Eduard Hoecherl, Konrad Wolf, Juergen Durner, Frank Klawonn, Stefan Holdenrieder

**Affiliations:** 1Munich Biomarker Research Center, Institute of Laboratory Medicine, German Heart Center, 80636 Munich, Germany; 2Institute of Clinical Chemistry, University Hospital Munich-Grosshadern, 81377 Munich, Germany; 3Leukocare AG, 81379 Munich, Germany; 4Department of Traumatology and Hand Surgery, Heinrich Heine University, 40225 Düsseldorf, Germany; 5Department of Trauma Surgery, München Klinik, 81545 Munich, Germany; 6Laboratory Becker MVZ GbR, 81671 Munich, Germany; 7Department of Conservative Dentistry and Periodontology, University Hospital, LMU, 80336 Munich, Germany; 8Department of Computer Science, Ostfalia University, 38302 Wolfenbüttel, Germany; 9Helmholtz Centre for Infection Research, 38124 Braunschweig, Germany

**Keywords:** cfDNA, plasma, trauma, prognosis, mortality, severity

## Abstract

**Background:** Trauma is still a major cause of mortality in people < 50 years of age. Biomarkers are needed to estimate the severity of the condition and the patient outcome. **Methods**: Cell-free DNA (cfDNA) and further laboratory markers were determined in plasma and serum of 164 patients at time of admission to the emergency room. Among them were 64 patients with severe trauma (Injury Severity Score (ISS) ≥ 16), 51 patients with moderate trauma (ISS < 16) and 49 patients with single fractures (24 femur neck and 25 ankle fractures). Disease severity was objectified by ISS and Glasgow Coma Scale (GCS). **Results:** cfDNA levels in plasma and serum were significantly higher in patients with severe multiple trauma (SMT) than in those with moderate trauma (*p* = 0.002, *p* = 0.003, respectively) or with single fractures (each *p* < 0.001). CfDNA in plasma and serum correlated very strongly with each other (R = 0.91; *p* < 0.001). The AUC in ROC curves for identification of SMT patients was 0.76 and 0.74 for cfDNA in plasma and serum, respectively—this was further increased to 0.84 by the combination of cfDNA and hemoglobin. Within the group of multiple trauma patients, cfDNA levels were significantly higher in more severely injured patients and patients with severe traumatic brain injury (GCS ≤ 8 versus GCS > 8). Thirteen (20.3%) of the multiple trauma patients died during the first week after trauma. Levels of cfDNA were significantly higher in non-surviving patients than in survivors (*p* < 0.001), reaching an AUC of 0.81 for cfDNA in both, plasma and serum, which was further increased by the combination with hemoglobin and leukocytes. **Conclusions:** cfDNA is valuable for estimation of trauma severity and prognosis of trauma patients.

## 1. Introduction

Trauma presents the leading cause of death for individuals aged 5 to 39 years in Germany and is a leading cause of death overall in all age groups worldwide [1]. Fifty percent of deaths occur within minutes; 20–30% die within several hours to 2 days after injury [2]. According to the German trauma registry (TR-DGU), 12% of patients in need of intensive care after trauma died in the hospital in 2021 [3]. The severity of trauma and the prognosis for survival and risk of complications can be estimated with the help of scores estimating anatomical lesions, such as the Injury Severity Score (ISS) [4] based on the Abbreviated Injury Scale (AIS 90) [5], or physiologic-based scores such as the Glasgow Coma Scale (GCS) [6] for the evaluation of the severity of traumatic brain injuries (TBI) [7].

Patients surviving the initial trauma frequently develop complications caused by posttraumatic immunologic changes [8,9]. Therefore, the early management of life-threatening complications within a short time and, hence, easily assessable and dependable biomarkers, are needed to estimate severity of the condition and possible resulting complications for the prognosis of the patient outcome.

Cell-free DNA (cfDNA) in the blood has recently gained increasing interest [10], as it is elevated in serum and plasma under physiological processes [11] such as pregnancy [12] and physical exercise [13] or during pathological processes, such as infections and sepsis [14], myocardial and other organ infarctions [15,16], autoimmune disorders [17], organ transplantation, thermal injuries [18,19], several types of cancers [20,21], surgery [20,22], as well as trauma [17,23]. The exact mechanism of cfDNA release from cells is still unclear [23]; however, apoptosis, necrosis, suicidal and vital NETosis with consecutive release of neutrophil extracellular traps (NETs) [24], erythroblast enucleation, phagocytosis and oncosis [25,26], as well as direct tissue damage from trauma are considered as potential sources [11,27]. As so-called danger-associated molecular patterns (DAMPs) [28], cfDNA is increased after traumatic injuries and plays a major role in the pathophysiology of the posttraumatic systemic inflammatory response commencing immediately after injury and contributing to post-injury complications [19,29,30]. The prediction of outcome in patients experiencing trauma in a variety of scenarios has been studied extensively [14,22,25,31,32,33], and cfDNA levels are associated with the severity of trauma and the prevalence of complications [20]. Due to the short half-life and stability of mono-nucleosomal cfDNA, from 15 min to 2 h [20,24,31], it could serve as a suitable marker in the critical emergency phase [20,31].

However, the assessment of cfDNA in qPCR methods is often time-consuming and laborious and, therefore, not suited for the emergency department. Here, we propose an easy-to-handle and quick method for cfDNA quantification in serum or plasma and show its potential—together with other lab-based markers—for the estimation of trauma severity and early hospital mortality in patients with multiple trauma, in comparison with already established routine laboratory biomarkers.

## 2. Materials and Methods

### 2.1. Patients

In this prospective study, 164 patients who were admitted to the Trauma Center at the Hospital Munich-Schwabing between September 2008 and October 2009 were enrolled. Exclusion criteria were admission later than 6 h after trauma, penetrating, thermal or chemical trauma, pathological fractures, neoplasms, and chronic inflammatory disease. Among the patients were 115 with multiple trauma who were admitted to the resuscitation room, additionally 24 patients with single femoral neck fractures (FNF) and 25 patients with single ankle fractures (AF) as controls. Characteristics of all patients are shown in Table 1.

The clinical assessment of the severity of trauma was objectified according to the ISS at admission to the hospital. Out of 115 patients with multiple trauma, 64 patients had severe multiple trauma (SMT; ISS ≥ 16) with a median ISS score of 29 (range: 16–75), and 51 patients had moderate multiple trauma (MMT; ISS < 16). Further, the GCS and the neurological status were assessed at admission and daily during the first week in all severely injured patients. Twenty-six patients suffered from severe TBI (GCS < 8), and 28 patients were admitted with mild to moderate TBI (GCS ≥ 8). Further, complications and mortality were recorded. Thirteen patients (20.3%) with severe polytrauma died within the first week.

The study was conducted in accordance with the ethical standards outlined in the Declaration of Helsinki and approved by the Ethics Committee of the Hospital of the Ludwigs-Maximilians-University (LMU) Munich (IRB nr. 405-07). Informed consents were obtained from the participants or a relative, if possible, during admission or following treatment. Postal inquiry for consent was conducted if the patient had already been released from the hospital. In case of no reply, the Ethics Committee of the LMU waived the requirement for approval, since the complete anonymization of the patient samples was given (IRB nr. 110-14, 29.09.2014).

### 2.2. Preanalytical and Analytical Methods

In addition to the routine clinical chemistry and hematologic parameter exams at time of admission to the resuscitation room, serum and EDTA plasma samples (Sarstedt, Nuermbrecht, Germany) were collected for cfDNA analyses. In case of minor trauma (AF, FNF) in control patients, blood samples were taken after the administration of diagnostic X-rays. The blood sample collection and preanalytical handling of blood samples until measurement were performed according to standard operating procedures (SOP) in cooperation with the Department of Clinical Chemistry of the Hospital Munich-Schwabing. Routine laboratory exams were performed in the Hospital Munich-Schwabing, including whole blood count with hemoglobin (HB), leukocytes and platelets, creatinine, creatine kinase, c-reactive protein (CRP), and liver enzymes. Additionally investigated parameters, High-Mobility-Group-Protein1 (HMGB1), soluble receptor for advanced glycation end products (sRAGE) and nucleosomes were analyzed as previously published [34].

For the analysis of cfDNA, samples were centrifuged at 3000× *g* for 15 min, and obtained serum and plasma were frozen locally at −20 °C on the same day and later transferred to the Biobank of the University Hospital Munich, where samples were stored at −80 °C until measurement. The quantification of serum and plasma cfDNA was conducted by use of a fluorescent Picogreen™ assay, according to the indications of the manufacturer (Leukocare, Munich, Germany). In brief, samples were centrifuged at 2000× *g* for 5 min before measurements. Additionally, four concentrations of DNA-standard samples (1500, 1000, 500 and 0 ng/mL) from calf thymus (Sigma, Taufkirchen, Germany) were measured in parallel, to establish a calibration curve. A 150 µL volume of phosphate-buffered saline (PBS) was added to 50 µL plasma/serum sample or 50 µL standard, followed by the addition of 150 µL diluted PicoGreen™ reagent (1 µL PicoGreen™ in 1 mL PBS). The fluorescence was measured by a fluorescence reader (Fusion; PerkinElmer; Monza, Italy) at 485 nm excitation and 530 nm emission wavelength. CfDNA concentrations were calculated using the calibration curve according to the specifications of the manufacturers.

Radiological assessment to objectify injuries consisted of computed tomography or X-ray in the emergency room. To estimate the seriousness of the pathological condition, clinical parameters and laboratory markers, as well as clinical scores were applied. The GCS for the assessment of severity of TBI and the ISS, as an anatomical overall score, were examined at time of admission in multiple injured patients.

### 2.3. Statistics

Descriptive statistics of the distribution of cfDNA concentrations in serum and plasma in the diverse patient groups are presented as median values and ranges and are illustrated as combined box- and dot-plots. Significance of biomarker differences in the defined subgroups in serum and plasma was assessed by Wilcoxon–Mann–Whitney test. Additionally, a principal component analysis (PCA) with all investigated parameters was conducted. The discriminative ability of the biomarker values for the severity of trauma and prognosis of first week hospital mortality was evaluated by means of receiver operating characteristics (ROC) curves and the respective areas under the curves (AUCs). In addition, a decision tree for the differentiation of trauma severity was calculated.

Correlations between cfDNA in serum and plasma and with further biomarkers were assessed by means of the Spearman rank coefficient, shown as a correlation plot. In multiple trauma patients, the associations of cfDNA with the severity of disease (by means of the ISS) and the severity of TBI (by means of the GCS) were also tested by the Spearman rank coefficient. Logistic regression analysis was carried out with forward and backward selection. Furthermore, random forest models were applied. The random forest model uses all predictors with an implicit selection of the most favorable markers. In addition, the random forest was validated with leave-one-out cross-validation. All comparisons were performed two-sided, and statistical significance was set at *p* < 0.05. Data analysis was performed using R (version 4.2.0; https://www.R-project.org (accessed on 12 February 2023), USA).

## 3. Results

### 3.1. Data Distribution of cfDNA in Trauma Groups

Considerable differences in cfDNA levels were observed between patients with SMT and patients with MMT or minor trauma (NFN, AF) in both serum and plasma, respectively. Generally, cfDNA levels were increased in all severely injured patients.

For cfDNA in plasma, levels were significantly higher in SMT patients (median plasma levels: 674 ng/mL) than in MMT patients (211 ng/mL; *p* = 0.002) or patients with FNF (172 ng/mL; *p* < 0.001) or AF (125 ng/mL; *p* < 0.001).

Similarly, cfDNA levels in serum were significantly higher in SMT patients (median serum level: 680 ng/mL) than in MMT patients (271 ng/mL; *p* = 0.003) or patients with FNF (203 ng/mL; *p* < 0.001) or AF (151 ng/mL; *p* < 0.001) (Figure 1; Table 2).

Among SMT patients, those with more severe TBI objectified by GCS ≤ 8 had significantly higher median levels in serum (1051 ng/mL) and plasma (1020 ng/mL) than patients with mild TBI and GCS > 8 (serum 525 ng/mL; *p* = 0.024; plasma 466 ng/mL; *p* = 0.046, respectively; Table 2).

Further, SMT patients who survived the first week of hospital stay had significantly lower median levels in serum (516 ng/mL) and plasma (470 ng/mL) as compared with patients who died during the first week in hospital (serum 1341 ng/mL; *p* < 0.001; plasma 1484 ng/mL; *p* < 0.001, respectively; Table 2).

For other laboratory parameters, significant differences between SMT and MMT were observed for creatinine kinase, creatinine (both *p* = 0.025), glucose (*p* = 0.011) and inversely for HB (*p* < 0.001), CRP (*p* = 0.008) and platelets (*p* = 0.045) (Appendix A).

If all parameters were joined in a PCA, patients with SMT and MMT could clearly be separated from the other groups with mild trauma (FNF, AF). Moreover, SMT and MMT patients showed different marker patterns as well (Figure 2).

### 3.2. Correlations of cfDNA and Hematology in Overall Patient Group

When correlating the investigated markers in the overall patient group, cfDNA in serum and plasma showed a strong correlation with each other (R = 0.91, *p* < 0.001). Furthermore, cfDNA moderately correlated inversely with HB (serum R = −0.34 (*p* = 0.002); plasma R = −0.36, (both *p* < 0.001)) and positively with leukocytes (serum R = 0.28 (*p* = 0.002); plasma R = 0.30 (*p* = <0.001)) but not with platelets. The correlation with other markers was not performed due to the number of missing values (Figure 3).

The distribution of hematology and further clinical chemistry parameters in SMT, MMT, FNF and AF is displayed in Appendix A. The correlation of plasma cfDNA with previously published [34] immunogenic cell death markers HMGB1 (R = 0.61), sRAGE (R = 0.39) and nucleosomes (R = 0.29) is shown in Appendix A.

### 3.3. Differential Diagnosis of Severe Multiple Trauma

For the distinction of patients with SMT from all other groups with trauma (MMT, FNF, AF), cfDNA in plasma and serum reached an AUC in ROC curves of 0.76 and 0.74, with sensitivities of 33% and 39% at 90% specificity, respectively. Among other lab markers, HB yielded the highest AUCs in ROC curves with 0.78 (HB inverse), with a sensitivity of 44% at 90% specificity (Figure 4A–C). When combining the most powerful markers in a linear regression model, the combination of cfDNA in plasma and HB provided the highest AUC with 0.84 and a sensitivity of 53% at 90% specificity (Figure 4D). Similar results were obtained in random forest analyses including all cfDNA and lab markers available (Figure 4E). Logistic regression was carried out with forward and backward selection, both leading to a model with only HB and cfDNA in plasma as predictors.

As a relevant clinical question, the distinction between patients with SMT and patients with MMT was investigated separately. For this comparison, cfDNA in serum and plasma reached an AUC in ROC curves of 0.66 and 0.67, respectively, with a sensitivity of 28% for both at 90% specificity. Among other lab markers, HB yielded the highest AUCs in ROC curves with 0.73 (HB inverse) and a sensitivity of 36% at 90% specificity (Figure 5A–C). When combining the most powerful markers in a logistic regression model, the combination of cfDNA in plasma and HB provided the highest AUC with 0.76 and a sensitivity of 48% at 90% specificity (Figure 5D). Similar results were obtained in random forest analyses including all cfDNA and lab markers available (Figure 5E).

Finally, for a more clinically practical application, a decision tree was calculated including HB, cfDNA in plasma, and platelets as the most relevant markers with defined cutoffs for identification of nodes with clear distinction between SMT, MMT, FNF and AF (Figure 6). In this stepwise analysis, it became evident that patients with low HB, high cfDNA and low platelet levels had the highest chance of having SMT, while patients with higher HB or those with low HB, low cfDNA and high platelet levels were more likely to be in the less severe groups. It has to be emphasized that other combinations achieved similar results, e.g., plasma cfDNA could have been exchanged with serum cfDNA.

### 3.4. Correlation and Prognosis of cfDNA and Trauma Severity

In patients with multiple SMT, high cfDNA levels in serum and plasma were correlated with the ISS for severity of disease by Spearman rank coefficient (R = 0.41, *p* < 0.001; R = 0.42, *p* = 0.001, respectively), as well as with the GCS score for severity of brain injury (R = −0.32, *p* = 0.02, both). If categorized, cfDNA levels were significantly higher in patients with severe TBI (GCS ≤ 8) in serum (median: 1051 ng/mL) and plasma (1020 ng/mL) than in patients with mild or moderate TBI (GCS > 8) in serum (525 ng/mL; *p* = 0.024) and plasma (466 ng/mL; *p* = 0.046) (Table 2, Figure 7A). The best discrimination of both groups was found for cfDNA in plasma (AUC: 0.66) and serum (AUC: 0.68) in ROC analysis, reaching 12% and 27% sensitivity at 90% specificity, respectively (Figure 7B,C). Logistic regression analysis lead to a model with the combination of cfDNA in plasma or serum with inverse HB and leukocytes (AUC 0.75) as predictors. Together, they achieved even higher AUCs as well as sensitivities of 42% at 90% specificity compared to single markers (Figure 7E). The random forest model for severity of TBI, including all parameters investigated, showed slightly less value.

### 3.5. Prognosis of In-Hospital First-Week Mortality

Within the group of patients with SMT, 13 patients (20.3%) died within the first week after trauma. Significantly higher values of cfDNA in serum were found in non-surviving (NS) (median: 1341 ng/mL) than in surviving (S) patients (516 ng/mL; *p* = 0.01). Similar results were obtained for cfDNA in plasma (median: 470 ng/mL for survivors and 1484 ng/mL for non-survivors, (*p* < 0.01)) (Figure 8A). The best discrimination of both groups was found for cfDNA in plasma (AUC: 0.81) and serum (AUC: 0.81) in ROC analysis, reaching 46% and 38% sensitivity at 90% specificity (Figure 8B,C). The best discriminative parameter was HB with an AUC of 0.78 and 38% sensitivity at 90% specificity (Figure 8D).

When combining the most powerful markers in a linear regression model, the combination of cfDNA in plasma or serum with HB and leukocytes achieved even higher AUCs of 0.88, as well as sensitivities of 62% at 90% specificity compared to single markers (Figure 8E). Similar results were obtained in random forest analyses including all cfDNA and lab markers available (Figure 8F). It has to be emphasized that other combinations achieved similar results, e.g., plasma cfDNA could have been exchanged with serum cfDNA.

## 4. Discussion

Despite the decline in mortality of SMT over the last few decades, trauma still is a leading cause of death throughout the world [1]. Patients surviving the initial trauma often are affected by a severe inflammatory response as a result of the release of DAMPs [29], which activate the innate and adaptive immunological response [19,35] via pattern recognition receptors (PRRs), followed by a local inflammatory response [29,36]. However, this activation could lead to a severe, uncontrolled systemic response, which can result in systemic inflammatory response syndrome (SIRS) [37] and ultimately in posttraumatic multi-organ failure (MOF) [38]. As an endogenous stress molecule or so-called alarmin [39,40] and part of DAMP, cfDNA in the blood stream, along with other DAMPs such as high mobility group box B protein, other cytokines or reactive oxygen species, is likely to trigger this mechanism [29,30]. Various mechanisms are discussed by which cfDNA is released from cells including damaged cells, dying or dead cells, as well as cells otherwise stimulated [23]—among others apoptosis, necrosis, suicidal and vital NETosis with consecutive release of neutrophil extracellular traps (NETs), erythroblast enucleation, phagocytosis, oncosis [25,26] and direct tissue damage from trauma [11,27,41]. cfDNA release into the blood stream may occur early after acute damage, and—despite the short half-life of 15 min to 2 h [20,24,31]—plasma levels may remain elevated for hours or days depending on the extent and dynamics of the trauma and the appearance of complications [41,42] in the further course.

Functional interaction with other DAMPs and immunogenic cell death (ICD) markers may play a role in immune activation, inflammation, and wound healing processes. Therefore, these markers may be informative for the outcome of patients suffering from multiple severe trauma.

The early information on prognosis for trauma patients at time of admission could have relevant therapeutic consequences: e.g., extended surgical intervention could be withheld in order to reduce the likelihood of second hits. Moreover, with the knowledge of the potential involvement of DAMPs such as cfDNA in the development of complications and increased mortality, new treatment options may result [23,32,43,44] and patient selection for the identification and prioritization of patients requiring extensive treatment may be improved, which could ultimately lead to a better outcome.

Therefore, we investigated cfDNA levels in the blood of trauma patients regarding their potential to predict severity and estimate prognosis as early as at time of admission to the hospital. In line with other studies, we confirmed that elevated cfDNA levels after trauma are clinically meaningful discriminative biomarkers for severity and prognosis [37,38]. Additionally, a review by Gögenur et al. [31], including 904 patients, confirmed its prognostic ability for the outcome in terms of mortality at time of admission, consistent with our study. With regard to the association of cfDNA with the severity of trauma, the authors reported inconsistent results. Although some studies have not found an association of cfDNA and trauma severity [45], others demonstrated a clear correlation between higher cfDNA levels in plasma and more severe trauma, e.g., objectified by the ISS [37,38] or other scores [32,46]. A recent study by Hazeldine et al. [32] reported significantly elevated concentrations of plasma cfDNA in TBI compared to healthy controls, less than one hour after injury. Patients developing multiple organ dysfunction syndrome also had significantly higher levels compared to patients with a non-eventful outcome. Similar findings were reported by Marcatti et al. [25], who showed higher amounts of plasma cfDNA in patients with TBI as compared with a control group using two different quantification methods, quantitative PCR and PicoGreen^TM^ staining.

In our study, patients with SMT could significantly be distinguished from moderately injured patients and those suffering from severe TBI (as the most important mortality factor after trauma [6]) from those with moderate or mild TBI by the level of measured cfDNA in serum and plasma, confirming the above mentioned results. Of particular interest was the significant correlation of cfDNA levels with the early hospital mortality. Within the group of severely injured patients, non-survivors had significantly higher cfDNA levels than survivors, in accordance with several studies [38,46]; the adverse prognosis could, for example lead to more invasive treatment choices to improve overall survival.

While confirming earlier prognostic results of cfDNA in trauma patients, the present study extended the approach by comparing cfDNA in plasma and in serum as well as by including other lab markers that are also assessed in routine settings.

Thereby, a high correlation between cfDNA in plasma and serum was found. This is remarkable, as often the superiority of plasma cfDNA over serum cfDNA is reported due to the additional release of DNA from stimulated cells during the coagulation process with subsequent higher cfDNA levels in serum [47,48,49]. Possibly the high concordance in trauma patients is explained by the fact that DNA is released not only by the injured tissue itself but also from maximally stimulated neutrophils that are activated as a result of trauma to protect the body from pathogen invasion and support wound healing after injury [41]. The process of NETosis is a well-known feature by which so called extracellular traps, consisting of disintegrated DNA and associated proteases, are ejected from neutrophils activated by damage and pathogen associated patterns (DAMPs and PAMPs) or platelet activation [41,50,51] within a short time to compartmentalize the injured region. It is speculated that a considerable part of the circulating cfDNA, particularly in patients with severe multiple trauma, derives from this massive process [26,52]. As major neutrophil stimulation is expected in both serum and plasma, the high correlation of resulting DNA levels in both materials is quite reasonable.

However, the source and mechanisms of release in trauma are still not conclusively resolved. Chornenki et al. [14] suggested that the source of elevated cfDNA levels after trauma is direct cell destruction resulting from trauma itself, differing from elevations in sepsis, which are likely released by activated neutrophils and resulting NETosis. They rejected the prognostic utility of cfDNA in trauma, due to lower levels in trauma patients compared to sepsis, even though citrullinated histone H3 and myeloperoxidase (MPO), as relevant markers of NETosis, were higher in trauma patients than in the control group. In line with this argumentation, Storz et al. [53] found that cfDNA indeed increased immediately after trauma but only weakly correlated with dramatic inflammatory and gene expression changes, rather serving as a biomarker for cellular stress or cell death and representing injury burden. On the other hand, Goswami et al. [52] identified increased H3 nucleosomal markers of NETosis in trauma patients. Further, Itagaki et al. [54] observed that circulating mitochondrial DNA released by trauma is an inducer of NETosis. It also is speculated that occlusion of small and medium vessels as a result of posttraumatic tissue swelling could result in higher shearing stress for circulating neutrophils with subsequent release of NETs. As seen in our study, there are obvious differences in cfDNA levels with regard to severity of trauma, and particularly in severe trauma, multiple mechanisms, tissue destruction and cell death, as well as activated neutrophils undergoing NETosis, seem to contribute to the resulting high levels of cfDNA in the blood circulation [26,52].

Previously published data analyzed the clinical role of other immunogenic cell death markers [34] such as HMGB1, sRAGE and nucleosomes. Interestingly, HMGB1 correlated especially well with cfDNA in plasma. Like cfDNA, HMGB1 is rapidly released after trauma [30,55] and elicits further immune-stimulatory effects after binding to sRAGE and toll-like receptors on macrophages, antigen presenting cells and dendritic cells, and also supports the induction of NET formation. Both cfDNA and HMGB1 could help in predicting trauma severity and mortality in trauma patients [34].

In addition, cfDNA correlates with several other laboratory parameters such as leukocytes and inversely with HB, as a sign of acute hemorrhage, and further with platelet number and creatinine kinase as a sign of muscle damage, which was particularly pronounced in the group with SMT. Interestingly, an elevation of inflammatory markers such as CRP and leukocytes was not evident at the very early time point at admission of severely traumatized patients in the emergency room, but CRP was significantly higher in patients with less severe injuries assessed at a later time point after the traumatic event (Appendix A). These observations demonstrate that plenty of processes are active in the acute phase after trauma, and cfDNA is one of the very early markers mirroring tissue damage and inflammation.

It must be emphasized that in this acute situation in the emergency room, fast and accurate diagnostics are essential to guide the treatment decision in the right direction. While laboratory parameters can be assessed quickly by modern laboratories in the hospital or by point-of-care devices on site, measurement of cfDNA levels in plasma and serum is often time-consuming and laborious. However, the results in the present study were obtained by an easy-to-handle, quick, low-cost, and robust method that offers the best conditions for such an emergency application.

Most remarkably, our results showed that the combination of cfDNA with other routine lab parameters, especially inverse HB and platelet levels, enhanced the differentiation between SMT and all other groups, between SMT and MMT as the relevant clinical control group, as well as between patients with high mortality risk during the first week after trauma and those with a more favorable outcome. Thereby, a clear additive value of these markers became evident. Thus, cfDNA is a highly valuable marker that adds to a better estimation of disease severity and prognosis in severe trauma patients at time of admission. Our results indicate that the severity of trauma is associated with the release cfDNA into the blood. The extent of tissue destruction and subsequent acute-phase reaction may both contribute to elevated cell death and cfDNA release. Therefore, cfDNA may be a relevant marker also in military trauma.

It is obvious that the present study has an explorative character, and results must be confirmed by independent prospective trials. Further limitations are the variability in patient presentation to the emergency room following major and minor injury and hence, the relatively long interval until samples were taken (up to six hours) after the event. Especially when considering the short half-life of cfDNA [20], variations in concentrations may result. However, this period was chosen to include patients with mild trauma, where blood samples were taken after initial imaging had taken place. In the group of severely injured patients, possible therapy applications, especially fluid administration prior to hospitalization, may have diluted cfDNA levels [35]. It also must be mentioned that other methods of cfDNA quantification, e.g., with different qPCR assays, may produce different results [11]. However, the fluorescent Picogreen™ staining method used here is known as a robust, reliable, accurate, quick and cost-efficient method. While all cfDNA irrespective of its origin is detected, it shows the sum effect of many pathological processes occurring after multiple traumata. Therefore, in this study, patients with minor impairments (AF, FNF) served as a control group, confirming the high relevance of cfDNA for the distinction of severity and for prognosis of polytraumatized patients. The study only examined patients with blunt trauma who arrived at the emergency room, to compare their cfDNA levels based on the severity of their condition. It can be speculated that cfDNA is also relevant in penetrating trauma. Additionally, because the focus was on the acute phase of trauma, other factors such as comorbidities and long-term outcomes were not investigated. Future investigations may address these areas to provide a more comprehensive understanding of the potential implications of cfDNA levels in trauma patients.

These shortcomings were balanced by a well-designed study setting that was close to clinical reality, defined time points of venipuncture, parallel routine laboratory analytics and professional documentation of clinical scores, standardized preanalytical handling of the study samples, high-quality analytics and independent statistics. The present results are a valuable basis for future prospective validation studies including the relevant markers and serial venipunctures at defined time points to learn about the individual post-traumatic kinetics of the markers and their prognostic implications.

## 5. Conclusions

cfDNA levels in serum and plasma are highly elevated in trauma and strongly associated with injury severity and poor prognosis of patients with multiple trauma. Therefore, cfDNA had additive value with routinely assessed blood parameters such as HB and platelet count. The practical clinical utility of cfDNA measurements is emphasized using an easy-to-handle, quick, reliable, and cost-efficient quantification assay.

## Figures and Tables

**Figure 1 diagnostics-13-01150-f001:**
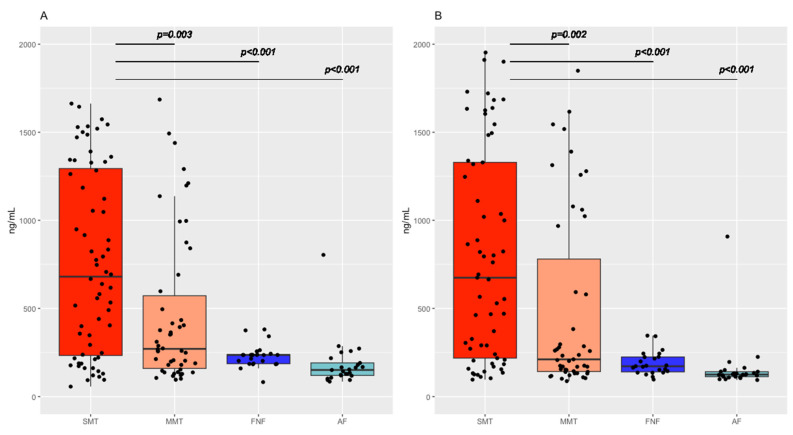
Boxplots for the distribution of cfDNA concentration in: (**A**) serum and (**B**) plasma in all groups investigated. Severe multiple trauma (SMT); moderate multiple trauma (MMT); femur neck fracture (FNF); ankle fracture (AF).

**Figure 2 diagnostics-13-01150-f002:**
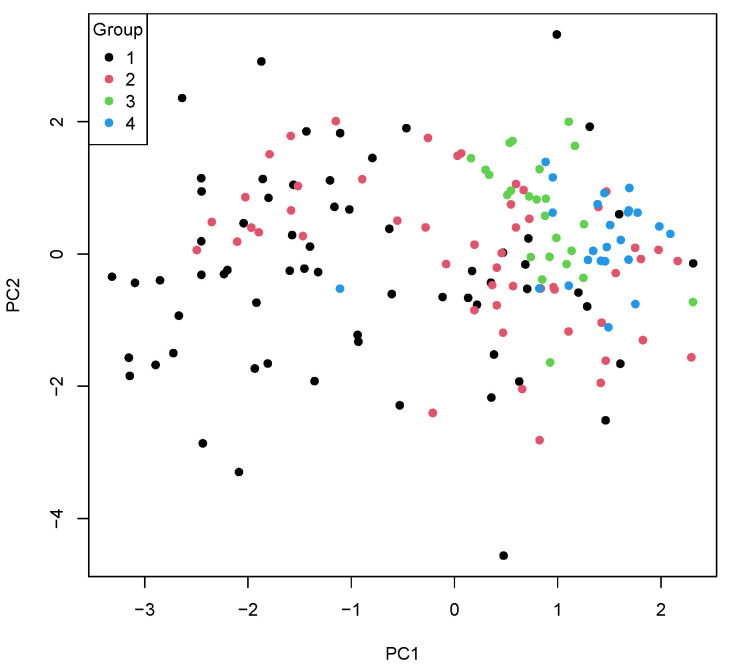
Principal component analysis (PCA) of all investigated parameters. Group 1 (black dots) = severe multiple trauma; group 2 (red dots) = moderate multiple trauma; group 3 (green dots) = femur neck fracture; group 4 (blue dots) = ankle fracture.

**Figure 3 diagnostics-13-01150-f003:**
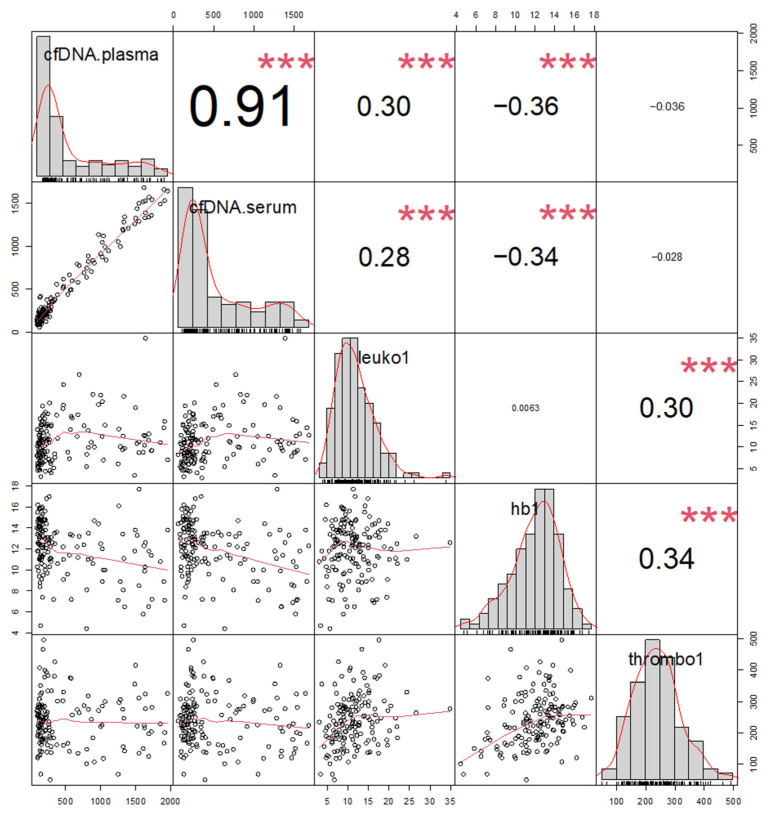
Correlation plots (graphics, lower left part) with corresponding correlation coefficients (numbers, upper right part) and value distribution of the markers (diagonal) for cfDNA in plasma and serum and hematology parameters leukocytes (Leuko1), hemoglobin (hb1), and platelets (thrombo1) for all patients. *** (*p* ≤ 0.001).

**Figure 4 diagnostics-13-01150-f004:**
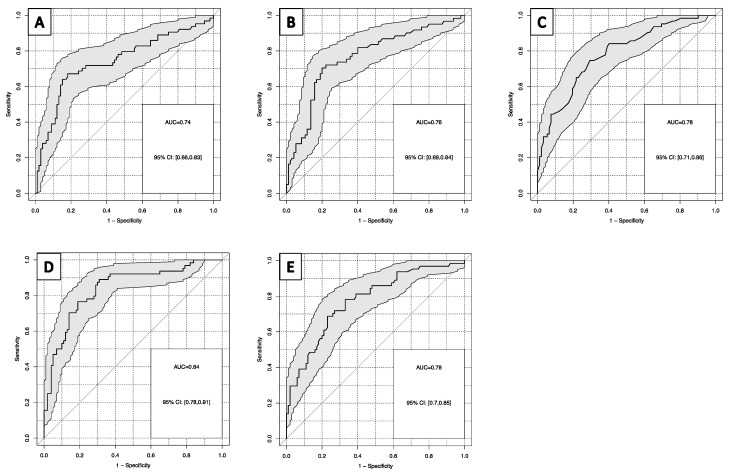
Receiver operating characteristic curves for the discrimination of severe multiple trauma and all other trauma groups for: (**A**) cfDNA in serum; (**B**) cfDNA in plasma; (**C**) hemoglobin; (**D**) hemoglobin and cfDNA in plasma; (**E**) random forest model of all lab parameters for severe multiple trauma versus other severity groups.

**Figure 5 diagnostics-13-01150-f005:**
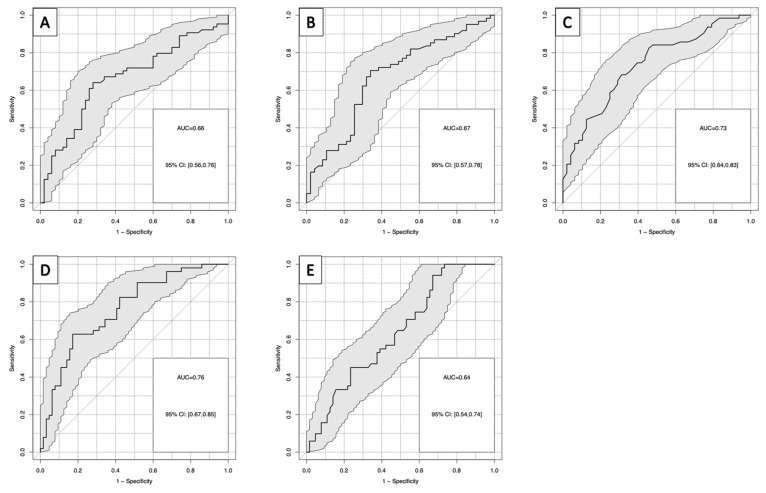
(**A**–**D**): Receiver operating characteristic curves for the discrimination of severe multiple trauma group and moderate trauma group for: (**A**) cfDNA in plasma; (**B**) cfDNA in serum; (**C**) hemoglobin; (**D**) hemoglobin and cfDNA in plasma; (**E**) random forest model of all investigated parameters for severe multiple trauma group versus moderate trauma group.

**Figure 6 diagnostics-13-01150-f006:**
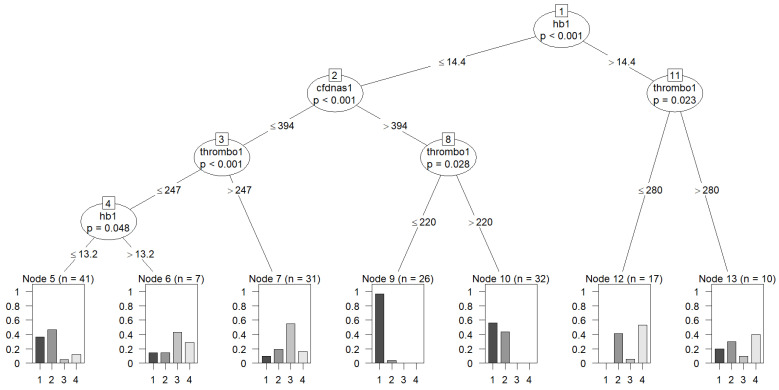
Decision tree for differentiation of all disease groups: group 1 = severe multiple trauma, group 2 = moderate multiple trauma; group 3 = femur neck fracture; group 4 = ankle fracture. hb1 (hemoglobin); cfdnas1 (cfDNA in serum); thrombo1 (platelets).

**Figure 7 diagnostics-13-01150-f007:**
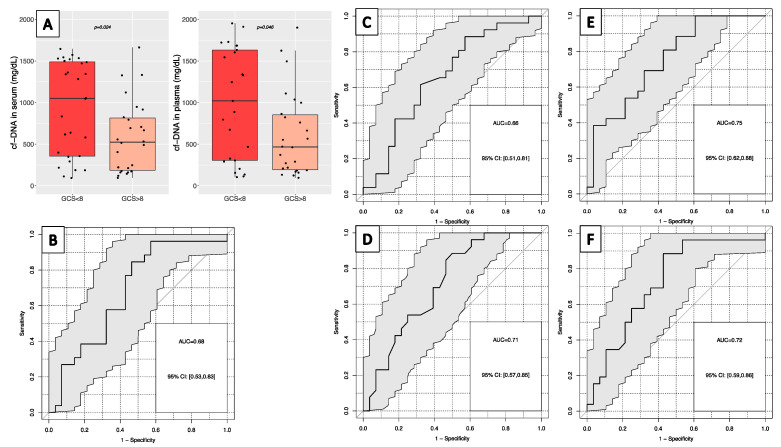
(**A**) Distribution of cfDNA levels in serum and plasma for severe traumatic brain injury (GCS ≤ 8) or mild/moderate brain injury (GCS > 8). Receiver operating characteristic curve for the discrimination of severity of traumatic brain injury for: (**B**) cfDNA in serum; (**C**) cfDNA in plasma; (**D**) hemoglobin; (**E**) cfDNA in serum, hemoglobin and leukocytes for the discrimination of severity of traumatic brain injury; (**F**) random forest model for traumatic brain injury severity.

**Figure 8 diagnostics-13-01150-f008:**
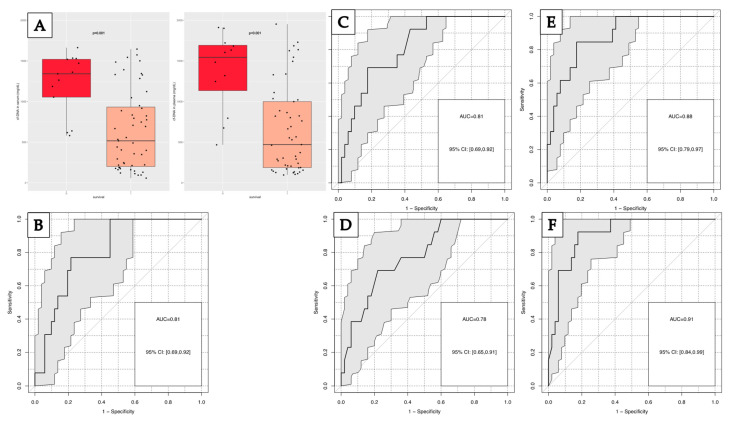
(**A**) Distribution of cfDNA concentration in serum and plasma in non-surviving (0) and surviving (1) patients; Receiver operating characteristic curve for the discrimination of one-week survival for: (**B**) cfDNA in serum; (**C**) cfDNA in plasma; (**D**) hemoglobin; (**E**) cfDNA in plasma, hemoglobin and leukocytes; (**F**) random forest model of all parameters for one-week mortality.

**Table 1 diagnostics-13-01150-t001:** Patient characteristics.

	Number of Patients	(%)	Age Median (Range)
I: Patients with severe multiple trauma (SMT, ISS ≥ 16)	64		43.4 (16–88)
Gender			
Female	24	37.5%	
Male	40	62.5%	
Head injury			
GCS ≤ 8	28	43.8%	
GCS > 8	26	40.6%	
No GCS available	10	15.6%	
Injured body region			
Head and neck	52	81.3%	
Chest	45	70.3%	
Abdomen	26	40.6%	
Face	28	43.8%	
Extremities	47	73.4%	
Survival of first week in hospital			
Yes	51	79.7%	
No	13	20.3%	
II: Patients with moderate multiple trauma (MMT, ISS < 16)	51		45.1 (16–93)
Gender			
Female	15	29.4%	
Male	36	70.6%	
III: Patients with femur fracture (FNF)	24		71.0 (32–86)
Gender			
Female	17	70.8%	
Male	7	29.2%	
IV: Patients with ankle fracture (AF)	25		50.7 (20–84)
Gender			
Female	10	40.0%	
Male	15	60.0%	

**Table 2 diagnostics-13-01150-t002:** cfDNA concentrations in serum and plasma in various disease groups. In patients with severe polytrauma, differences in cfDNA concentrations in severe versus moderate TBI and in first week survival and non-survival are shown.

	N	cfDNA Serum (ng/mL)		cfDNA Plasma (ng/mL)	
		Median	Range	Median	Range
SMT	64	680	57–1663	674	96–1953
MMT	51	271	96–1686	211	88–1849
*p-value*		*p = 0.003*		*p = 0.002*	
FNF	24	203	82–387	172	97–346
*p-value*		*p < 0.001*		*p < 0.001*	
AF	25	151	85–804	125	94–908
*p-value*		*p < 0.001*		*p < 0.001*	
TBI in SMT					
GCS ≤ 8	28	1051	93 -1645	1020	104–1953
GCS > 8	26	525	95–1663	466	96–1901
*p-value*		*p = 0.024*		*p = 0.046*	
Survival in SMT					
yes	51	516	57–1645	470	96 -1953
no	13	1341	581–1663	1484	467–1911
*p-value*		*p = 0.001*		*p < 0.001*	

## Data Availability

Not applicable.

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
