# Peer review of "Cell-Free DNA in Plasma and Serum Indicates Disease Severity and Prognosis in Blunt Trauma Patients"

_diagnostics, 2023, doi:10.3390/diagnostics13061150_

Round 1
Reviewer 1 Report
-Title: The study refers only to blunt trauma patients and not in trauma patients collectively
-It is not justified why penetrating trauma is excluded. The same study must concern penetrating trauma also as another study arm.
-Thermal/Chemical injuries probably excluded also? it is not clarified in the exclusion criteria
-Table 1: in the "injured body region" the term external refers to what?
survival of first week is presented only for the patients with SMT
-Materials and Methods, first paragraph, line8 (114), must be corrected
-What about cfDNA levels and morbidity ?
-what about penetrating trauma and thermal injuries? Why not studying cfDNA levels in these groups also?
-Any recommendations for future studies in military trauma as well which has deferent mechanisms?
-references might be slightly fewer
Reviewer 2 Report
Journal: Diagnostics (ISSN 2075-4418)
Manuscript ID: diagnostics-2244470
Type: Article
Title:
Cell-free DNA in plasma and serum indicates disease severity and prognosis in trauma patients
Authors:
Inga Maria Trulson , Juliane Stahl , Stefan Margraf , Martin Scholz , Eduard Höcherl , Konrad Wolf , Jürgen Durner , Frank Klawonn , Stefan Holdenrieder *
Section:
Pathology and Molecular Diagnostics
Special Issue
Cell-Free Nucleic Acids—New Insights into Physico-Chemical Properties, Analytical Considerations, and Clinical Applications
Aim of the study: Authors propose an easy-to-handle and quick method for cfDNA quantification in serum or plasma and show their potential together with other lab-based markers for the estimation of trauma severity and early hospital mortality in patients with multiple trauma, in comparison with already established routine laboratory biomarkers.
Ø Results
Line 212: Figure 3: Correlation plot needs more clarification to be simpler for readers
Line 264: NFN: Do you mean femur neck fracture, FNF?
Line 278: R=0.41, p<0.001; R=0.42, p=0.001……….kindly revise p values
Fig.7.A and Fig.8.A……. The x & y axis titles should be more obvious
Line 288: sensitivities of 42 at 90% specificity………..42%
Ø Discussion
Line 334-338: Different mechanism of cfDNA release from damaged, dying or stimulated cells are discussed……Reformulate this sentence
Line 338: [11,28] [45]: Do you mean [11,28,45]
Line 378: the adverse prognosis could e.g. result in more invasive treatment choices……………what do you mean by could e.g. result ?
Ø Limitations of the study, conclusion and recommendations are all fulfilled and discussed
